# Predictive Model of Gemtuzumab Ozogamicin Response in Childhood Acute Myeloid Leukemia on Event-Free Survival: Data Analysis Based on Trial AAML0531

**DOI:** 10.3390/bioengineering12030297

**Published:** 2025-03-14

**Authors:** Kun-Yin Qiu, Xiong-Yu Liao, Jian-Pei Fang, Dun-Hua Zhou

**Affiliations:** 1Department of Hematology/Oncology, Children’s Medical Center, Sun Yat-sen Memorial Hospital, Sun Yat-sen University, Guangzhou 510120, China; qiuky@mail2.sysu.edu.cn (K.-Y.Q.); liaoxy7@mail.sysu.edu.cn (X.-Y.L.); 2Guangdong Provincial Key Laboratory of Malignant Tumor Epigenetics and Gene Regulation, Sun Yat-sen Memorial Hospital, Sun Yat-sen University, Guangzhou 510120, China

**Keywords:** gemtuzumab ozogamicin, children, acute myeloid leukemia, trial AAML0531, predictive model

## Abstract

**Purpose:** We aimed to develop a simple nomogram and online calculator that can identify the optimal subpopulation of pediatric acute myeloid leukemia (AML) patients who would benefit most from gemtuzumab ozogamicin (GO) therapy. **Methods:** Within the framework of the phase Ⅲ AAML0531 randomized trial for GO, the event-free survival (EFS) probability was calculated using a predictor-based nomogram to evaluate GO treatment impact on EFS in relation to baseline characteristics. Nomogram performance was assessed by the area under the receiver operating characteristic curve (AUC) and the calibration curve with 500 bootstrap resample validations. Decision curve analysis (DCA) was performed to evaluate the clinical utility of the nomogram. **Results:** A total of 705 patients were randomly assigned to two arms: the No-GO arm (n = 358) and the GO arm (n = 347). We performed a nomogram model for EFS among childhood AML. The AUC (C statistic) of the nomogram was 0.731 (95%CI: 0.614–0.762) in the development group and 0.700 (95% CI: 0.506–0.889) in the validation group. DCA showed that the model in the development and validation groups had a net benefit when the risk thresholds were 0–0.75 and 0–0.75, respectively. Notably, an intriguing observation emerged wherein pediatric patients with AML exhibited a favorable outcome in the GO arm when the predicted 5-year EFS probability fell below 60%, demonstrating a superior EFS compared to the No-GO Arm. **Conclusions:** We have developed a nomogram and online calculator that can be used to predict EFS among childhood AML based on trial AAML0531, and this might help deciding which patients can benefit from GO.

## 1. Introduction

Acute myeloid leukemia (AML) is a heterogeneous disorder characterized by unfavorable prognosis, accounting for 15–20% of childhood leukemias. While treatment outcomes have improved over the past decades, approximately 30–40% of pediatric AML patients still experience relapse, highlighting the need for novel therapeutic approaches [1,2,3]. Childhood AML exhibits distinct biological characteristics compared to adult AML, including higher frequencies of core-binding factor translocations (e.g., t(8;21) and inv(16)) and lower incidence of adverse genetic markers like FLT3-ITD, which necessitate age-specific treatment strategies [4,5,6].

The cell-surface antigen CD33 is expressed in over 90% of pediatric AML cases, with expression patterns showing developmental stage-dependent variations that differ from adult populations [7,8,9]. Gemtuzumab ozogamicin (GO), a humanized anti-CD33 antibody conjugated to calicheamicin, has shown particular promise in pediatric AML. Initial phase II studies (AAML00P2 and AAML03P1) demonstrated the safety and efficacy of GO combined with chemotherapy in children, achieving 3-year event-free survival (EFS) rates of 53–63% [10,11,12]. The Children’s Oncology Group (COG) AAML0531 phase III trial, the largest randomized study specifically designed for pediatric AML, established that GO addition improved 3-year EFS from 46.9% to 53.1% (*p* = 0.04), though this benefit was not uniformly distributed across all risk groups [13].

Despite these advances, childhood AML presents unique challenges in treatment optimization. The developing hematopoietic system’s increased sensitivity to chemotherapy toxicity, combined with the long-term sequelae of intensive therapy in growing children, underscores the need for precision medicine approaches [14,15,16]. Current risk stratification systems, primarily based on cytogenetics and treatment response, fail to adequately predict which patients will benefit most from targeted therapies like GO. This limitation is particularly relevant given the recent FDA reapproval of GO for CD33-positive AML in 2017, which included pediatric indications but lacked age-specific guidance [17,18,19].

A critical knowledge gap persists in identifying the optimal pediatric subpopulation for GO therapy. While meta-analyses of adult trials suggest particular benefit in favorable-risk AML, pediatric studies have shown conflicting results regarding the interaction between molecular subtypes and GO response [20,21,22]. Furthermore, the long-term follow-up data from AAML0531 (median 4.9 years) revealed diminishing GO benefits over time, emphasizing the need for robust predictive tools specific to childhood AML [23,24,25,26,27,28].

This study addresses a critical gap in pediatric AML management by developing the first GO response prediction model specifically for childhood AML, integrating molecular markers (WT1/CEBPA) with clinical parameters. Our web-based nomogram represents a novel precision medicine tool that outperforms existing risk stratification systems in identifying GO-responsive subgroups.

## 2. Patients and Methods

### 2.1. Patients

Pediatric patients newly diagnosed with AML who participated in the COG AAML0531 phase III trial, spanning from August 2006 to June 2010, were eligible for the present analysis. The findings presented herein are derived, in whole or in part, from the data generated by the Therapeutically Applicable Research to Generate Effective Treatments (TARGET) initiative (https://ocg.cancer.gov/programs/target (accessible on 1 January 2025)) under the identifier phs000218. The data utilized for this analysis can be accessed at https://portal.gdc.cancer.gov/projects (accessible on 1 January 2025). Notably, the primary distinction between this study and prior investigations lies in its exclusive enrollment of pediatric patients with AML, specifically those under the age of 18 years.

The demographic characteristics were compared between the No-GO and GO arms of the study. Inclusion criteria included newly diagnosed de novo AML, centralized cytogenetic/molecular profiling, and full treatment adherence per protocol. Key exclusions included acute promyelocytic leukemia (APL) or prior chemotherapy. A total of 705 patients with de novo acute myeloid leukemia (AML), aged 0 to 18 years, were randomly allocated to either the no-GO arm (n = 358) or the GO arm (n = 347). Of the entire study population, 51.5% (363 patients) were male and 48.5% (342 patients) were female, with a median age of 10 years. The randomization process resulted in well-balanced study arms, with no significant differences observed between the two arms in terms of gender, age, white blood cell count (WBC), the French–American–British (FAB) category, peripheral blood (PB) blast count, bone marrow (BM) blast count, karyotype, risk group, or gene mutation status.

After each induction course, remission was evaluated thoroughly. Upon completion of IND1, there were comparable rates of CR, not-in-CR status, and death. At the conclusion of IND2, no statistically significant difference in CR rates was observed between the two arms. In summary, the overall induction remission rates were similar across both arms, indicating that the two arms were comparable in this regard. The demographic information of 705 eligible pediatric patients with AML, as classified by the GO, is presented in Table 1.

### 2.2. Study Design

In total, 705 patients diagnosed with de novo AML, within the age range of 0 to 18 years, were randomly assigned to two distinct treatment arms: the standard five-course chemotherapy arm (designated as the No-GO arm; comprising 358 patients) and the same chemotherapy regimen augmented with two doses of GO at 3 mg/m^2^ per dose (designated as the GO arm; consisting of 347 patients). All patients were treated under the Children’s Oncology Group (COG) AAML0531 phase III trial (NCT00372593), which mandated a fixed chemotherapy backbone for all participants. No modifications to chemotherapeutic agents, dosing, or scheduling occurred during the trial period (2006–2014). Protocol amendments were restricted to supportive care (e.g., antimicrobial prophylaxis), with no changes to the chemotherapy backbone.

This allocation was carried out during the first induction course (IND1) and the second intensification phase (as detailed in Appendix A). Prior to the administration of GO, chemotherapy-induced cytoreduction was performed to optimize CD33 target saturation, rather than escalating the dosage of GO. Concurrent administration of anthracyclines was avoided to mitigate the potential for additive hepatotoxicity. Risk stratification within both treatment arms guided the decision to proceed with SCT, based on diagnostic molecular/cytogenetic risk factors and the disease response observed after IND1, as specified below.

Low risk (LR) was defined based on the presence of t(8;21)(q22;q22), inv(16)(p13.1q22), or t(16;16)(p13.1;q22). Patients with LR were not considered for SCT. High risk (HR) was determined by the presence of monosomy 7, monosomy 5/5q deletion, or persistent disease (PD) with bone marrow blasts exceeding 15% at the end of IND1. Following enrollment of 374 eligible patients, the presence of FLT-3 internal tandem duplication with a high allelic ratio (>0.4; FLT3-ITD HAR) was added as a criterion for HR classification. Cytogenetics was the primary determinant in risk classification, with FLT3-ITD HAR overriding favorable cytogenetics. All HR patients underwent best allogeneic SCT (nonsyngeneic matched family donor or unrelated donor) following INT1 or INT3. HR patients without suitable donors continued with their assigned chemotherapy regimen. Standard risk (SR) was defined by the absence of low- or high-risk factors, and these patients only received SCT if a matched family donor was available. Patients allocated to SCT underwent the procedure after INT1. Consequently, patients randomly assigned to the GO arm received only one dose during IND1. Response classification was determined by morphologic examination of bone marrow blasts, with complete remission (CR) defined as less than 5% blasts, partial remission (PR) as 5% to 15% blasts, and PD as more than 15% blasts. Patients with refractory disease (RD) were excluded from the protocol therapy. RD was defined as the presence of central nervous system (CNS) disease after IND1, bone marrow blasts ≥ 5%, or any extramedullary disease at the end of IND2. The clinical protocol was approved by the institutional review boards of all participating institutions, and the COG Myeloid Disease Biology Committee approved the research study. The trial was conducted in accordance with the Declaration of Helsinki and was registered at www.clinicaltrials.gov under NCT00372593.

### 2.3. Cytogenetic and Molecular Profiling

Centralized analysis was performed at COG-certified laboratories using conventional karyotyping and FISH for core abnormalities (t(8;21), inv(16), 11q23/MLL rearrangements), standard techniques in pediatric AML during the 2000s. FLT3-ITD status was determined via fragment analysis (sensitivity: 5% mutant alleles), consistent with era-specific guidelines.

### 2.4. Treatment Response and Follow-Up

Patients were deemed to be in CR if they exhibited less than 5% blasts and the absence of extramedullary disease following a single course of induction chemotherapy. Minimal residual disease (MRD) was determined using flow cytometry and was considered positive if 0.1% or more of the disease was detected at the conclusion of induction I. Clinical outcome data for patients enrolled in the COG AAML051 trial were analyzed as of 30 September 2014. The median follow-up period for eligible patients who were alive at the time of last contact and included in our analysis was 4.9 years (ranging from 0 to 9.6 years). EFS was defined as the duration from study entry until the occurrence of death, induction failure, or any type of relapse. Patients who were lost to follow-up were censored at the date of their last known contact.

### 2.5. Statistical Analyses

Univariable analyses were performed to identify prognostic factors associated with event-free survival (EFS). Categorical variables (e.g., sex, FAB subtype) were compared using Chi-square test or Fisher’s exact test as appropriate. Continuous variables (e.g., WBC, age) were analyzed with Mann–Whitney U test or Kruskal–Wallis test based on data distribution, using forward stepwise selection with Akaike information criterion. Variables reaching *p* < 0.05 were included in the multivariable Cox proportional hazards model. The final prediction model was developed using three core components: 1. Discrimination: ROC curve analysis (AUC) evaluating model’s ability to distinguish high/low-risk patients. 2. Calibration: bootstrap-corrected curves comparing predicted vs. observed 5-year EFS probabilities. 3. Clinical utility: decision curve analysis quantifying net benefit across clinical decision thresholds. Internal validation was performed using 1000 bootstrap resamples to prevent overfitting. All analyses were conducted with R 4.2.2 (survival and rms packages).

## 3. Results

### 3.1. Univariable and Multivariable Analyses

In the analysis, risk factors that were deemed significant during univariable analysis were incorporated into multivariable models, with the aim of elucidating the independent effects of these factors on EFS among pediatric patients with AML, as outlined in Table 2. Through multivariate analysis, it was determined that WBC; (*p* < 0.001) and WT1 mutation (*p* < 0.001) emerged as independent factors that have a detrimental impact on EFS. Conversely, favorable effects on EFS for the entire cohort were observed in association with FAB M7 subtype (*p* = 0.017), inv(16) cytogenetic abnormality (*p* = 0.014), t(8;21) translocation (*p* = 0.008), and CEBPA mutation (*p* = 0.011). Notably, no significant association was detected between GO treatment and EFS (*p* = 0.818). Additionally, a comparable 5-year EFS rate was observed between patients who did not receive GO treatment and those who did (44.8% versus 45.5%, respectively; *p* = 0.7; refer to Appendix A for further details).

### 3.2. Development and Assessment of the Predictive Nomogram

#### 3.2.1. Model Development

Given the absence of a discernible difference in 5-year EFS between the No-GO and GO arms, we have undertaken further research to develop a predictive model aimed at identifying the optimal patient population for the GO arms. To achieve this, subjects were randomly assigned to either the development group (n = 529; 75%) or the validation group. Initially, utilizing the finalized multivariable model, we created a nomogram. This was accomplished by assigning a weighted score to each factor associated with 5-year EFS in pediatric AML. Each predictor was assigned a specific score on a rating scale, and the total points for each variable were summed. A vertical line was then drawn downward from the total points to indicate the corresponding probability of EFS. Notably, a higher score corresponded to a lower probability of EFS, as depicted in Figure 1.

The area under the curve (AUC) of the receiver operating characteristic (ROC) curve for this comprehensive nomogram was determined to be 0.731, with a 95%CI ranging from 0.614 to 0.762 (Appendix A). Subsequent to 1000 bootstrap iterations, the calibration curves demonstrated a satisfactory alignment between the observed and predicted probabilities within the nomogram, thereby attesting to the stability and precision of the predictive models (Appendix A). Furthermore, the curves revealed that the model yielded a net benefit when the risk threshold was situated within the range of 0 to 0.75 (Appendix A).

#### 3.2.2. Model Validation

To evaluate the internal validity of the model, it was implemented on the remaining 25% of the randomly selected internal validation dataset. The assessment of internal validity in childhood AML patients was conducted utilizing the C-statistic, calibration, and decision curve analysis. The results of the 25% random internal validation indicated a commendable predictive performance and stability of the nomogram models. Specifically, the AUC (C-statistic) of the model within the validation cohort was 0.700 (95% CI: 0.506–0.889), as depicted in Appendix A. Additionally, the nomogram calibration curve of the validation cohort demonstrated satisfactory calibration, as shown in Appendix A. The decision curves further revealed that the model yielded a net benefit when the risk threshold ranged from 0 to 0.75, as illustrated in Appendix A.

### 3.3. Survival Analysis for the Entire Cohort

As depicted in the aforementioned nomogram models, the identified independent predictors were subsequently employed to estimate the 5-year EFS probability for patients in both the No-GO and GO arms. Notably, an intriguing observation emerged wherein pediatric patients with AML exhibited a favorable outcome in the GO arm when the predicted 5-year EFS probability fell below 60%, demonstrating a superior EFS compared to the No-GO Arm. Conversely, this advantage dissipated in the GO arm patients when the predicted EFS probability surpassed 60%, accompanied by a marked decline in EFS (Figure 2). The statistical significance of the interaction between the predicted EFS and the GO arm treatment arm was established (*p* < 0.05), underscoring the variability in the GO arm’s therapeutic benefit contingent upon the predicted EFS. Subsequent to bootstrap resampling, the significance of the interaction term persisted (*p* < 0.05). Importantly, when the predicted EFS probability exceeded 60%, no discernible difference was observed between the two arms (*p* > 0.05). Furthermore, we have developed an online calculator accessible at http://www.empowerstats.net/pmodel/?m=11688_PredictiveModelofGemtuzumabOzogamicinResponse, accessed on 25 January 2025 (Figure 3), which facilitates the prediction of 5-year EFS among pediatric AML patients, thereby aiding pediatricians in making informed decisions regarding the initiation of GO treatment.

## 4. Discussion

Based on several randomized trials, the recent US Food and Drug Administration approval of GO, a CD33-directed agent, represents significant advancements in AML treatment. Nevertheless, the efficacy of GO in pediatric AML remains a matter of debate. The TARGET program, a collaboration between the COG and the National Cancer Institute (NCI), offers a robust platform for studying pediatric cancer cohorts [29]. Consequently, we accessed the TARGET dataset to retrieve trial AAML0531 data. The AAML0531 trial, the largest randomized pediatric de novo AML trial to date and the sole pediatric randomized controlled trial, to incorporate GO into induction and intensification, demonstrated that GO significantly enhanced EFS at the 3-year mark (53.1% versus 46.9%; *p* = 0.04) [26]. However, our analysis within the context of the COG AAML0531 trial revealed no statistically significant improvement in 5-year EFS with GO (*p* = 0.7). These discrepancies with prior research can be attributed to our extended follow-up period and our study’s exclusive focus on patients under the age of 18. Given the lack of difference in 5-year EFS between the No-GO and GO arms, we have developed a predictive model aimed at identifying the optimal population for GO administration.

Previous research has conclusively demonstrated that EFS in pediatric AML is not merely a single risk factor, but rather a complex interplay of numerous factors [30,31]. The nomogram’s incorporation of WBC and the WT1 mutation aligns with prior pediatric AML studies, where hyperleukocytosis and WT1 aberrations correlate with chemoresistance and relapse. High WBC and WT1 mutations identified subgroups deriving significant GO benefit. Mechanistically, elevated WBC may reflect CD33-high blast populations susceptible to GO, while WT1 mutations are linked to chemoresistance pathways counteracted by GO’s DNA damage mechanism. Conversely, CEBPA mutations and inv(16) predicted favorable outcomes independent of GO, suggesting limited added value in these subgroups.

Importantly, the lack of GO benefit in high-EFS subgroups suggests that CD33-independent pathways may dominate in biologically favorable AML, warranting further investigation. Notably, our current study reveals that elevated WBC levels and the presence of the WT1 mutation are predictive of poorer EFS in childhood AML, whereas CEBPA mutation, as well as the presence of inv(16) and t(8;21) karyotypic abnormalities, are indicative of a more favorable prognosis. These findings are in line with previous investigations, emphasizing the strong correlation between these factors and EFS in pediatric AML [31,32,33].

In the current study, we have devised and rigorously tested a nomogram to predict EFS among pediatric patients with AML. This predictive model is based on several key factors: WBC, FAB category, karyotype, WT1 mutation status, and CEBPA mutation status. The AUC values obtained in the training and validation cohorts were 0.731 and 0.700, respectively, underscoring the excellent accuracy and robustness of the nomogram. The usability of this model is enhanced by the limited number of predictors required and the straightforwardness of the computational procedures, making it a comprehensive and sensitive tool for identifying pediatric AML patients with a poor prognosis.

Our nomogram model is capable of deriving the cumulative score of risk factors and the probability of EFS, thereby facilitating pediatricians in providing more pertinent recommendations to patients diagnosed with AML. Notably, prior research endeavors have also developed predictive models aimed at forecasting EFS among pediatric AML cases [34,35]. Nevertheless, these prior models were constructed primarily based on data sourced from single-center studies or limited sample sizes. Consequently, they may be constrained by the absence of a comprehensive dataset derived from a multi-center, randomized phase Ⅲ trial akin to the COG AAML0531 study. In contrast, our model exhibits enhanced precision and credibility in comparison to these previous studies.

In our comprehensive assessment, we have deemed it necessary to meticulously balance the potential advantages of GO treatment against its individual-level drawbacks. Consequently, our current report offers invaluable insights to the existing literature, carrying significant clinical ramifications. Specifically, we have leveraged an online calculator to determine an individual’s predicted EFS, thereby facilitating an accurate decision on the suitability of GO treatment. For instance, if the nomogram’s total score corresponds to a predicted EFS of 70%, the EFS for the No-GO arm surpasses that of the GO arm, but the difference is not significant. Based on risk-benefit assessment, we advise the patient to opt for the No-GO arm. Conversely, if the nomogram predicts an EFS of 50% and the EFS for the GO arm is higher than that of the No-GO arm, the patient should undergo GO treatment. This approach empowers pediatricians and patients to make well-informed clinical decisions when evaluating the role of GO treatment in AML.

We acknowledge the fact that our study has some limitations. First of all, our study did not incorporate CD33 expression levels, a critical determinant of GO efficacy, due to data unavailability in the TARGET cohort. Future validation in pediatric cohorts with CD33 quantification (e.g., flow cytometry, transcriptomics) is essential to refine predictive accuracy. Secondly, the prediction model of this study was only internally validated and not externally validated. Thirdly, retrospective analysis of biomarker data introduces potential selection bias, though all samples were prospectively collected. Future work should do the following: (1) validate the model in multicenter cohorts; (2) incorporate emerging biomarkers (e.g., FLT3-ITD allelic ratio); (3) assess cost-effectiveness in resource-limited settings.

In conclusion, our nomogram and online calculator are simple to use and able to predict the EFS among childhood AML based on trial AAML0531. The novel nomogram total score corresponds to predicted 5-year EFS, and this might help deciding which patients can benefit from GO treatment, so as to achieve the purpose of precise treatment of pediatric AML.

## Figures and Tables

**Figure 1 bioengineering-12-00297-f001:**
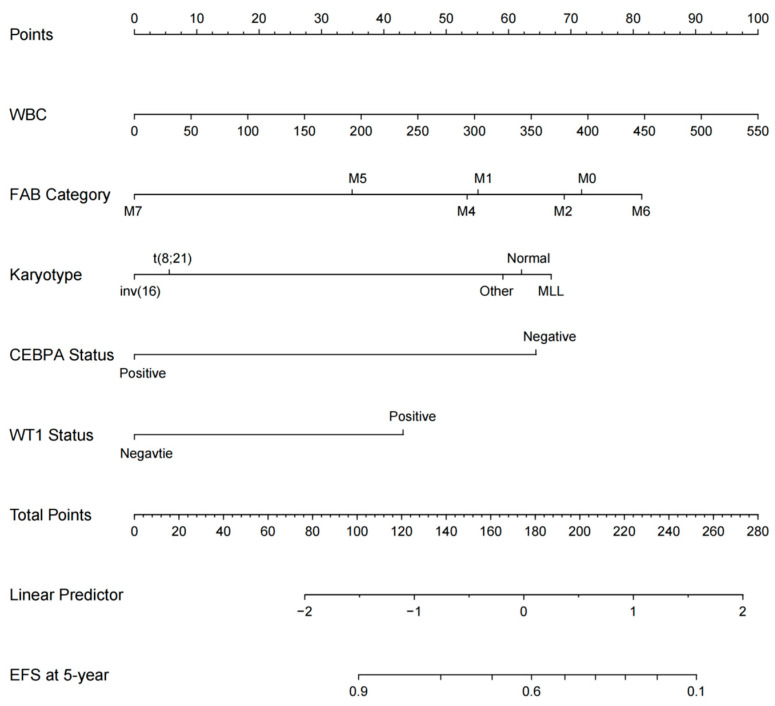
Nomogram to predict the probability of 5-year EFS in childhood AML patients.

**Figure 2 bioengineering-12-00297-f002:**
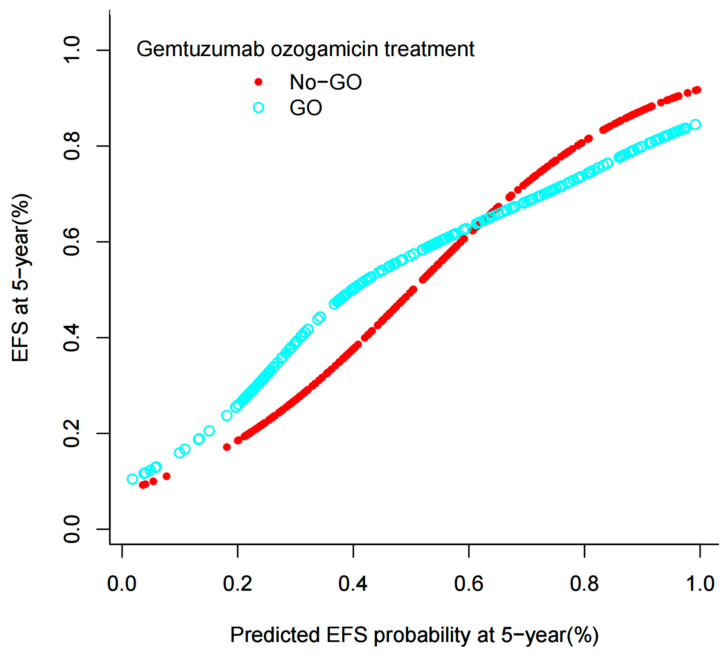
EFS rate plotted against predicted probability of EFS at 5-year after diagnosis. The lines for No-GO and GO cross at 60%.

**Figure 3 bioengineering-12-00297-f003:**
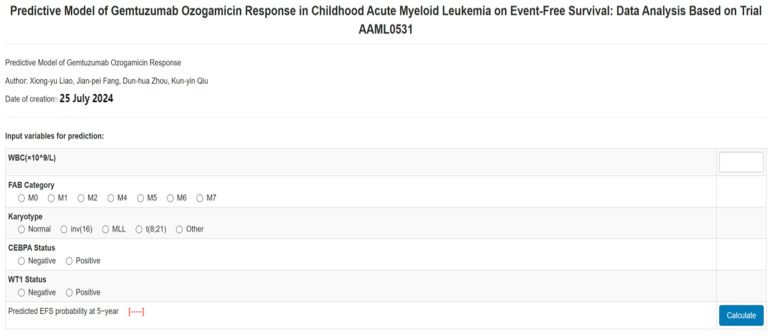
The online calculation for predicting probability of EFS at 5-year among childhood AML.

**Table 1 bioengineering-12-00297-t001:** Baseline characteristics of study participants by GO status classification.

Characteristics	Total	GO Status		*p* Value
		No-GO (n = 358)	GO (n = 347)	
**Gender, n (%)**				0.248
Male	363 (51.5%)	192 (53.6%)	171 (49.3%)	
Female	342 (48.5%)	166 (46.4%)	176 (50.7%)	
Age (y), median (range)	10.0 (0.0–18.0)	10.2 (0.0–18.0)	9.7 (0.0–17.9)	0.442
FAB category				0.798
M0	24 (4.1%)	12 (4.2%)	12 (4.0%)	
M1	78 (13.3%)	42 (14.6%)	36 (12.0%)	
M2	156 (26.6%)	77 (26.8%)	79 (26.3%)	
M4	161 (27.4%)	78 (27.2%)	83 (27.7%)	
M5	128 (21.8%)	56 (19.5%)	72 (24.0%)	
M6	13 (2.2%)	8 (2.8%)	5 (1.7%)	
M7	27 (4.6%)	14 (4.9%)	13 (4.3%)	
Initial WBC (×10^9^/L), median (range)	33.5 (0.2–519.0)	35.3 (0.2–473.1)	32.7 (0.8–519.0)	0.612
PB blast (%)	48.0 (0.0–98.0)	47.0 (0.0–98.0)	49.0 (0.0–98.0)	0.645
BM blast (%)	71.0 (0.0–100.0)	71.0 (0.0–99.0)	71.0 (3.0–100.0)	0.331
Karyotype				0.526
Normal	167 (24.3%)	81 (23.1%)	86 (25.6%)	
inv(16)	91 (13.3%)	46 (13.1%)	45 (13.4%)	
MLL	125 (18.2%)	62 (17.7%)	63 (18.8%)	
t(8;21)	109 (15.9%)	64 (18.3%)	45 (13.4%)	
Other	194 (28.3%)	97 (27.7%)	97 (28.9%)	
Risk group, n (%)				0.568
Low risk	262 (38.2%)	140 (40.1%)	122 (36.2%)	
Standard risk	305 (44.5%)	151 (43.3%)	154 (45.7%)	
High risk	119 (17.3%)	58 (16.6%)	61 (18.1%)	
CNSL, n (%)				0.541
No	654 (92.8%)	330 (92.2%)	324 (93.4%)	
Yes	51 (7.2%)	28 (7.8%)	23 (6.6%)	
FLT3-ITD status, n (%)				0.879
FLT3-ITD wild-type	555 (78.7%)	281 (78.5%)	274 (79.0%)	
FLT3-ITD mutation	150 (21.3%)	77 (21.5%)	73 (21.0%)	
CEBPA status, n (%)				0.898
CEBPA wild-type	1743 (94.5%)	342 (96.6%)	323 (94.4%)	
CEBPA mutation	101 (5.5%)	12 (3.4%)	19 (5.6%)	
NPM1 status, n (%)				0.480
NPM1 wild-type	639 (91.4%)	321 (90.7%)	318 (92.2%)	
NPM1 mutation	60 (8.6%)	33 (9.3%)	27 (7.8%)	
WT1 status, n (%)				0.923
WT1 wild-type	647 (92.6%)	328 (92.7%)	319 (92.5%)	
WT1 mutation	52 (7.4%)	26 (7.3%)	26 (7.5%)	
End of IND1 response				0.103
CR	513 (73.6%)	257 (72.2%)	256 (75.1%)	
Not in CR	176 (25.3%)	92 (25.8%)	84 (24.6%)	
Death	8 (1.1%)	7 (2.0%)	1 (0.3%)	
End of IND2 response				0.099
CR	600 (86.8%)	300 (84.5%)	300 (89.3%)	
Not in CR	79 (11.4%)	46 (13.0%)	33 (9.8%)	
Death	12 (1.7%)	9 (2.5%)	3 (0.9%)	
SCT in first CR				0.633
No	518 (81.8%)	265 (82.6%)	253 (81.1%)	
Yes	115 (18.2%)	56 (17.4%)	59 (18.9%)	

**Abbreviations:** WBC, white blood cell counts; PB blast, peripheral blood blast; BM, bone marrow blast; CNSL, central nervous system leukemia; CEBPA, CCAAT/enhancer-binding protein alpha; FLT3-ITD, fms-related tyrosine kinase 3; NPM1, nucleophosmin 1; WT1, Wilms tumor 1; CR, complete remission; IND, induction course; SCT, stem cell transplantation.

**Table 2 bioengineering-12-00297-t002:** Univariable and multivariable analyses on EFS.

Variables	Univariable Analysis		Multivariable Analysis	
	*HR* (95%CI)	*p* Value	*HR* (95%CI)	*p* Value
Gender				
Male	Ref.		Ref.	
Female	0.9 (0.7, 1.1)	0.143	0.8 (0.6, 1.0)	0.107
Age	1.0 (1.0, 1.0)	0.995	1.0 (1.0, 1.0)	0.084
Risk group				
Low risk	Ref.		Ref.	
Standard risk	3.1 (2.4, 4.0)	<0.001	1.7 (0.7, 3.9)	0.225
High risk	3.3 (2.4, 4.4)	<0.001	1.1 (0.5, 2.6)	0.800
WBC	1.0 (1.0, 1.0)	<0.001	1.0 (1.0, 1.0)	<0.001
BM blast (%)	1.0 (1.0, 1.0)	<0.001	1.0 (1.0, 1.0)	0.109
Peripheral blasts (%)	1.0 (1.0, 1.0)	0.092	1.0 (1.0, 1.0)	0.989
CNSL				
No	Ref.		Ref.	
Yes	0.9 (0.6, 1.4)	0.649	1.2 (0.8, 1.9)	0.437
FAB category				
M0	Ref.		Ref.	
M1	0.5 (0.3, 0.9)	0.012	0.8 (0.4, 1.4)	0.418
M2	0.3 (0.2, 0.6)	<0.001	0.9 (0.5, 1.8)	0.854
M4	0.5 (0.3, 0.8)	0.005	0.9 (0.5, 1.7)	0.804
M5	0.6 (0.4, 0.9)	0.024	0.6 (0.3, 1.0)	0.065
M6	0.8 (0.3, 1.7)	0.497	0.7 (0.3, 1.7)	0.416
M7	0.1 (0.1, 0.4)	<0.001	0.3 (0.1, 0.8)	0.017
Karyotype				
Normal	Ref.		Ref.	
inv(16)	0.5 (0.3, 0.8)	<0.001	0.3 (0.1, 0.8)	0.014
MLL	1.2 (0.9, 1.7)	0.138	0.8 (0.5, 1.3)	0.316
t(8;21)	0.4 (0.3, 0.6)	<0.001	0.3 (0.1, 0.7)	0.008
Other	1.2 (0.9, 1.6)	0.122	0.8 (0.5, 1.1)	0.190
FLT3-ITD status				
FLT3-ITD wild-type	Ref.		Ref.	
FLT3-ITD mutation	1.6 (1.2, 2.0)	<0.001	1.0 (0.6, 1.5)	0.875
NPM1 status				
NPM1 wild-type	Ref.		Ref.	
NPM1 mutation	0.5 (0.3, 0.8)	0.003	0.5 (0.2, 1.0)	0.055
CEBPA status				
CEBPA wild-type	Ref.		Ref.	
CEBPA mutation	0.4 (0.2, 0.8)	0.008	0.2 (0.1, 0.7)	0.011
WT1 status				
WT1 wild-type	Ref.		Ref.	
WT1 mutation	2.3 (1.7, 3.2)	<0.001	2.1 (1.4, 3.2)	<0.001
Gemtuzumab ozogamicin treatment				
No-GO	Ref.		Ref.	
GO	0.9 (0.8, 1.1)	0.538	1.0 (0.8, 1.3)	0.818

**Abbreviations:** WBC, white blood cell counts; BM, bone marrow blast; CNSL, central nervous system leukemia; CEBPA, CCAAT/enhancer-binding protein alpha; FLT3-ITD, fms-related tyrosine kinase 3; NPM1, nucleophosmin 1; WT1, Wilms tumor 1.

## Data Availability

The data in the current study are available from the corresponding author on reasonable request.

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
