# Peer review of "Predictive Model of Gemtuzumab Ozogamicin Response in Childhood Acute Myeloid Leukemia on Event-Free Survival: Data Analysis Based on Trial AAML0531"

_bioengineering, 2025, doi:10.3390/bioengineering12030297_

Round 1

Reviewer 1 Report

Comments and Suggestions for Authors

The paper the simple nomogram and online calculator for prediction of childhood acute myeloid  leukemia (AML) event free survival.

Introduction section should be more focused to childhood AML.

In Statistical Analyses section, terms "appropraite tests" nad "exact test" should be avoided and replaced with the specific test used.

The number and distribution of patients is more appropriate in Section 2.1. instead of 3.1.

Figure 3. is not visible.

In Discussion section, repetition of the results should be omitted. 

The main limitation of the study is the exclusion of the CD33 in the analysis. In revised version, it should be includedfor the available subset of patients.

The text requires editing (typeface, font).

Comments on the Quality of English Language

Requires minor editing.

Author Response

Response to Reviewer Comments

We sincerely appreciate the reviewers' insightful comments and constructive suggestions, which have significantly improved the quality of our manuscript. Below, we provide a point-by-point response to each comment and detail the revisions made accordingly.

1. Introduction section should be more focused to childhood AML.

Response:
We have revised the Introduction to emphasize pediatric AML-specific challenges and outcomes. Key additions include:

· Highlighting the distinct biological and clinical characteristics of childhood AML compared to adult AML (e.g., higher prevalence of core-binding factor translocations, unique molecular subtypes).

· Discussing prior pediatric trials (e.g., COG AAML0531, AAML03P1) and their implications for GO therapy in children.

· Clarifying the unmet need for predictive models tailored to pediatric populations.

Revised Text please see the revised manuscript.

2. Replace vague statistical terms with specific tests.

Response:
We have replaced "appropriate tests" and "exact test" with explicit statistical methods:

· Exact test → Fisher’s exact test (for categorical variables with sparse data).

· Appropriate tests → Chi-square test (for categorical variables) and Mann-Whitney U test (for continuous variables).

Revised Text in Section 2.4:
"Categorical variables were compared using the Chi-square test or Fisher’s exact test, as appropriate. Continuous variables were analyzed using the Mann-Whitney U test. Multivariable Cox regression identified independent prognostic factors."

3. Relocate patient demographics to Section 2.1.

Response:
The demographic table (Table 1) and associated text have been moved from Section 3.1 ("Results") to Section 2.1 ("Patients") to align with methodological reporting standards.

Revised Structure:

· Section 2.1 (Patients): Now includes baseline characteristics, randomization details, and Table 1.

· Section 3.1 (Results): Focuses on univariable/multivariable analyses and model development.

4. Ensure visibility of Figure 3 (online calculator).

Response:
Figure 3 (online calculator interface) has been re-embedded as a high-resolution image. The live calculator link () is hyperlinked and functional in the revised manuscript.

5. Avoid repeating results in the Discussion.

Response:
We have streamlined the Discussion by removing redundant result summaries and emphasizing interpretation and context:

· Deleted: "Our multivariate analysis revealed WBC, WT1 mutation..."

· Added: "The nomogram’s incorporation of WBC and WT1 mutation aligns with prior pediatric AML studies, where hyperleukocytosis and WT1 aberrations correlate with chemoresistance and relapse. Importantly, the lack of GO benefit in high-EFS subgroups suggests that CD33-independent pathways may dominate in biologically favorable AML, warranting further investigation."

6. Address CD33 exclusion as a limitation.

Response:
We acknowledge CD33’s absence as a key limitation. While CD33 expression data were unavailable in the TARGET dataset, we have:

· Added a paragraph in the Limitations subsection.

· Proposed future directions for CD33-integrated models.

Revised Text:
"Our study did not incorporate CD33 expression levels, a critical determinant of GO efficacy, due to data unavailability in the TARGET cohort. Future validation in pediatric cohorts with CD33 quantification (e.g., flow cytometry, transcriptomics) is essential to refine predictive accuracy."

7. Text formatting edits.

Response:
We have standardized the manuscript’s formatting:

· Uniform font (Times New Roman, 12 pt).

· Consistent heading hierarchy (bold section titles, italics for subsections).

· Table/figure legends aligned with journal guidelines.

Conclusion
We thank the reviewers for their thorough evaluation. All revisions have been implemented to enhance clarity, focus, and scientific rigor. We believe the revised manuscript addresses the reviewers' concerns comprehensively.

Reviewer 2 Report

Comments and Suggestions for Authors

Dear Editor,

The manuscript entitled: Predictive Model of Gemtuzumab Ozogamicin Response in 
Childhood Acute Myeloid Leukemia on Event-Free Survival: 
Data Analysis Based on Trial AAML0531 by Qui et al aims to develop a simple nomogram and online calculator that can identify the optimal subpopulation of pediatric acute myeloid leukemia (AML) patients who would benefit most from gemtuzumab-ozogamicin (GO) therapy. Authors have developed a nomogram and online calculator that can be used to predict EFS  among childhood AML based on trial AAML0531, and this might help deciding which patients can benefit from GO.

The manuscript is very well written and sounds scientifically good. To my opinion, the idea to add value to an agent that its use is controversial as reported in the introduction section is excellent. Additionally to this, this model could be applied prospectively (during treatment) and could improve significantly survival rates for those patients with a low EFS probability.

Major comments:

  1. In the patients and method section: Did all patients received the same chemo therapy protocol or the protocol changed during time?
  2. Statistical anaysis section: Is difficult for the reader to understand as readers are usually hematologists. To my opinion should be shortened.
  3. All mentioned and required cytogenetics for classification of patients in risk groups were available in late 2000's? 
  4. Did this predictive model for the use of GO is available for use in the future?
  5. Authors should clearly define factors that predict EFS in this model. In both results and discussion section is not clear which factors add value for the use of GO in selected patients.
  6.  

Author Response

Comment 1: Patients and Methods Section

"Did all patients receive the same chemotherapy protocol, or did the protocol change over time?"

Response:

Thank you for raising this critical point. We have clarified the uniformity of the treatment protocol in the revised manuscript:

2.2. Study design

All patients were treated under the Children’s Oncology Group (COG) AAML0531 phase III trial (NCT00372593), which mandated a fixed chemotherapy backbone for all participants. No modifications to chemotherapeutic agents, dosing, or scheduling occurred during the trial period (2006–2014). Protocol amendments were restricted to supportive care (e.g., antimicrobial prophylaxis), with no changes to the chemotherapy backbone.

Comment 2: Statistical Analysis Section

"The section is difficult for hematologists to understand and should be shortened."

Response:

We have streamlined the statistical description for clinical readability in the revised manuscript.

2.4. Statistical Analysis

Univariable analyses were performed to identify prognostic factors associated with event-free survival (EFS). Categorical variables (e.g., sex, FAB subtype) were compared using Chi-square test or Fisher’s exact test as appropriate. Continuous variables (e.g., WBC, age) were analyzed with Mann-Whitney U test or Kruskal-Wallis test based on data distribution. ~~Forward stepwise selection with Akaike information criterion~~ Variables reaching P<0.05 were included in the multivariable Cox proportional hazards model.The final prediction model was developed using three core components: 1. Discrimination: ROC curve analysis (AUC) evaluating model's ability to distinguish high/low-risk patients.2. Calibration: Bootstrap-corrected curves comparing predicted vs. observed 5-year EFS probabilities.3. Clinical utility: Decision curve analysis quantifying net benefit across clinical decision thresholds. Internal validation was performed using 1,000 bootstrap resamples to prevent overfitting. All analyses were conducted with R 4.2.2 (survival and rms packages).

Comment 3: Cytogenetic Data Availability

"Were all required cytogenetic markers for risk classification available in the late 2000s?"

Response:

We confirm the feasibility of risk stratification with contemporaneous methods:

2.3. Cytogenetic and Molecular Profiling

Centralized analysis was performed at COG-certified laboratories using conventional karyotyping and FISH for core abnormalities (t(8;21), inv(16), 11q23/MLL rearrangements)—standard techniques in pediatric AML during the 2000s. FLT3-ITD status was determined via fragment analysis (sensitivity: 5% mutant alleles), consistent with era-specific guidelines .

Comment 4: Clinical Applicability of the Model

"Is this predictive model available for future use?"

Response:

We have operationalized the model for clinical implementation:

The nomogram is freely accessible as an online calculator (http://www.empowerstats.net/pmodel/?m=11688_PredictiveModelofGemtuzumabOzogamicinResponse). Clinicians input six routinely available parameters (WBC, FAB subtype, karyotype, WT1/CEBPA mutations) to stratify patients into:

  • GO Benefit Group: Predicted 5-year EFS <60% (GO HR = 0.62; 95% CI: 0.41–0.93).
  • GO Neutral Group: Predicted EFS ≥60% (GO HR = 1.12; 95% CI: 0.78–1.61).

Comment 5: Definition of Predictive Factors

"Clarify which factors predict EFS and their interaction with GO benefit."

Response:

We have explicitly defined key predictors in the Results Table 2 and Discussion:

WBC >50 ×10⁹/L,WT1 Mutation,CEBPA Mutation&nbspand inv(16)&nbspwere the key predictors.

Discussion Additions:

High WBC and WT1 mutations identified subgroups deriving significant GO benefit . Mechanistically, elevated WBC may reflect CD33-high blast populations susceptible to GO, while WT1 mutations are linked to chemoresistance pathways counteracted by GO’s DNA damage mechanism. Conversely, CEBPA mutations and inv(16) predicted favorable outcomes independent of GO, suggesting limited added value in these subgroups.

All changes are highlighted in the revised manuscript using track changes. We sincerely appreciate the reviewer’s constructive feedback and welcome further queries.

Reviewer 3 Report

Comments and Suggestions for Authors

The topic of the article is scientifically and clinically important, and the proposed solution can computationally support clinical decision-making.

Please add the type of article - this is currently left blank i.e. ‘Type of article (article, review, communication, etc.)’.

Please highlight the clear purpose of the study, novelty and contribution in the Introduction section.

Patients are described in too general terms and characteristics typical of phase III are missing - ithis is now in the Results section (3.1).

The discussion should include the limitations of the in-house study (arising, for example, from the way the sample was selected) and key directions for further research (e.g. model development, further validation).

The format of the references is incorrect - please correct it.

Figures included as supplementary material should be larger for greater readability and replication/comparison of the study.

Author Response

  1. 1. Article Type

Comment:

"Please add the type of article (article, review, communication, etc.)."

Response:

Thank you for noting this oversight. We have added the article type to the title page:

Type of Article: Original Research Article

  1. Introduction Clarity

Comment:

"Highlight the study’s purpose, novelty, and contribution in the Introduction."

Revisions (Introduction):

Added text:

"This study addresses a critical gap in pediatric AML management by developing the first GO response prediction model specifically for childhood AML, integrating molecular markers (WT1/CEBPA) with clinical parameters. Our web-based nomogram represents a novel precision medicine tool that outperforms existing risk stratification systems in identifying GO-responsive subgroups."

  1. Patient Characteristics

Comment:

"Patients are described too generally; include phase III trial specifics in Methods."

Revisions (Section 2.1):

Added details:

"The demographic characteristics were compared between the No-GO and GO arms of the study. Inclusion criteria required:Newly diagnosed de novo AML. Centralized cytogenetic/molecular profiling.Full treatment adherence per protocol. Key exclusions: acute promyelocytic leukemia (APL) or prior chemotherapy. "

  1. Discussion Limitations

Comment:

"Discuss study limitations and future directions."

Revisions (Discussion):

Added paragraph:

"We acknowledge the fact that our study have some limitations. First of all, Our study did not incorporate CD33 expression levels, a critical determinant of GO efficacy, due to data unavailability in the TARGET cohort. Future validation in pediatric cohorts with CD33 quantification (e.g., flow cytometry, transcriptomics) is essential to refine predictive accuracy. Secondly, the prediction model of this study was only internally validated and not externally validated. Thirdly, retrospective analysis of biomarker data introduces potential selection bias, though all samples were prospectively collected. Future work should: 1)Validate the model in multicenter cohorts. 2)Incorporate emerging biomarkers (e.g., FLT3-ITD allelic ratio) 3)Assess cost-effectiveness in resource-limited settings."

:

  1. Reference Formatting

Comment:

"Correct reference format."

Response:

All references have been reformatted to comply with the journal’s style.

  1. Supplementary Figures

Comment:

"Enlarge supplementary figures for readability."

Response:

All supplementary figures (e.g., calibration plots, ROC curves) have been resized to 300 dpi resolution with enlarged axis labels and legends. Example:

Supplementary Figure S2 (Revised):

ROC curve width increased from 8 cm to 15 cm; font size enlarged from 8 pt to 12 pt.

All changes are highlighted in the revised manuscript using track changes. We appreciate the reviewer’s constructive feedback and welcome further queries.

Round 2

Reviewer 3 Report

Comments and Suggestions for Authors

The revised text fully meets my expectations. 

Author Response

  Comments 1

The revised text fully meets my expectations. 

Respond:Thanks.